# SPARSE CONCEPT ANCHORING FOR INTERPRETABLE AND CONTROLLABLE NEURAL REPRESENTATIONS

**Sandy Fraser**
Independent Researcher
Melbourne, Australia
Alexander.Fraser@alumni.anu.edu.au

**Patryk Wielopolski**
Independent Researcher
Wrocław, Poland

## ABSTRACT

We introduce **Sparse Concept Anchoring**, a method that biases latent space to position a targeted subset of concepts while allowing others to self-organize, using only minimal supervision (in our setting, labels for $< 0.1\%$ of examples per anchored concept). Training combines activation normalization, a separation regularizer, and anchor or subspace regularizers that attract rare labeled examples to predefined directions or axis-aligned subspaces. The anchored geometry enables two practical interventions: reversible behavioral steering that projects out a concept's latent component at inference, and permanent removal via targeted weight ablation of anchored dimensions. Experiments on structured autoencoders show selective attenuation of targeted concepts with negligible impact on orthogonal features, and complete elimination with reconstruction error approaching theoretical bounds. Sparse Concept Anchoring therefore provides a practical pathway to interpretable, steerable behavior in learned representations.

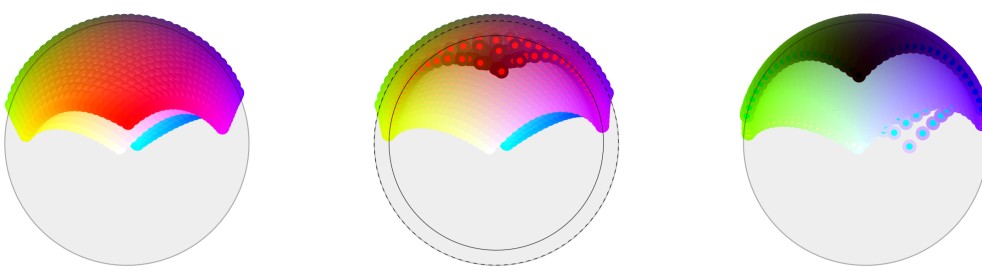

Figure 1: **Sparse Concept Anchoring organizes latent space predictably using minimal supervision, enabling behavioral steering and permanent concept deletion.** *Left*: Supervision on *red* during training organizes related concepts around the anchor point. *Center*: The resulting structure enables behavioral steering, demonstrated here by repelling *red* stimuli toward nearby colors while preserving other model capabilities. *Right*: Permanent concept deletion removes *red* responses entirely via weight ablation.

## 1 INTRODUCTION

As AI systems grow more capable, predicting, understanding, and controlling their internal representations has become a critical challenge for safety and interpretability (Bereska & Gavves, 2024). Existing approaches navigate trade-offs between supervision requirements, architectural constraints, and intervention reliability. Concept Bottleneck Models achieve strong interpretability by predicting concepts as intermediate representations but require full concept supervision. Recent variants reduce supervision through sparsity and unsupervised discovery, while post-hoc methods such as Concept Activation Vectors (Kim et al., 2018) and steering vectors (Rimsky et al., 2024) preserve model flexibility but face reliability challenges when concept directions do not align with emergent geometry. Indeed, post-hoc explanations of black-box models cannot be fully faithful to the original computation (Rudin, 2019).

Building on sparse concept learning (Semenov et al., 2024; Sawada & Nakamura, 2022; Oikarinen et al., 2023; Yamaguchi et al., 2025) and geometric representation learning using hypersphere constraints (Wang & Isola, 2020; Loshchilov et al., 2024), we introduce a framework combining minimal supervision with explicit geometric separation to induce interpretable representations of *specific concepts*. This approach is advantageous for four reasons. First, neural networks lose plasticity early in training and establish connectivity patterns during critical periods that become difficult to reshape later (Achille et al., 2019), favoring up-front over post-hoc approaches. Second, related concepts naturally cluster in latent space (Bengio et al., 2013; Mikolov et al., 2013; Pennington et al., 2014), and explicit geometric constraints on the hypersphere can complement this natural organization. Third, localized concept representations enable reliable targeted interventions: concepts occupying known, separable locations can be selectively removed without relying on the model's emergent geometry. Finally, if safety-critical concepts constitute a small fraction of the diverse knowledge encoded in large models—as suggested by the millions of features discovered in Claude 3 Sonnet, of which only a small subset relate to safety concerns (Templeton et al., 2024)—then anchoring a handful of such concepts should not interfere with general capability acquisition.

We propose **Sparse Concept Anchoring** (SCA), a framework that fixes selected concepts to predetermined locations in latent space using minimal supervision during training. Our approach combines task-specific loss with targeted regularization to embed specific concepts while preserving representational flexibility. The framework introduces two complementary inductive biases: *structural constraints* across all data points to shape global latent geometry, and *concept organizational regularizers* applied only to samples associated with anchored concepts. This organization enables two intervention classes: *behavioral steering*, which dynamically modifies activations, and *permanent concept removal* through weight ablation.

To validate the approach, we use color reconstruction as a controlled testbed with well-defined concept relationships and interpretable latent geometry. Primary colors provide distinct conceptual anchors, secondary colors emerge from their combinations, and the resulting structure can be visualized directly.[1] Using autoencoders trained on RGB data, we demonstrate that anchoring a single concept (*red*) with noisy supervision on only $83 \pm 8 \approx 0.09\%$ labeled examples out of 96,064 training samples induces sufficient latent organization to enable reliable behavioral steering and targeted concept removal, reducing red channel reconstruction error to near-theoretical limits while preserving reconstruction of orthogonal colors.

In summary, our contributions are:

- We introduce a framework that inverts traditional interpretability workflows by establishing predictable concept locations during training rather than discovering them post-hoc, using labels for $< 0.1\%$ of training examples per concept in our experiments.

- We demonstrate two mechanistically distinct intervention classes on these anchored representations: reversible behavioral steering via activation projection and permanent concept removal via weight ablation, both operating without post-hoc analysis.

- We validate the approach on color reconstruction, achieving selective concept attenuation with reconstruction error approaching theoretical bounds while preserving orthogonal features.

## 2 SPARSE CONCEPT ANCHORING WITH MINIMAL SUPERVISION

Sparse Concept Anchoring selectively anchors a limited set of concepts using minimal supervision during training. It integrates two complementary inductive biases alongside the primary task objective: *structural constraints* that enforce global latent geometry, and *concept-organizational regularizers* that position specific concepts. This selective shaping preserves representational flexibility while ensuring predictable locations for key concepts.

---

[1]We shall discuss color spaces, but we make no measurement or claim relating to human perception of color.

## 2.1 PROBLEM FORMULATION AND ARCHITECTURE

We consider a dataset $\mathbf{X} = \{\boldsymbol{x}^{(i)}\}_{i=1}^{N}$ of $N$ samples, where each $D$-dimensional vector $\boldsymbol{x}^{(i)} \in \mathbb{R}^D$ represents the input space (e.g., RGB color values with $D = 3$). We denote $\mathcal{C} = \{c_1, c_2, \ldots, c_K\}$ as the set of $K$ concepts to be anchored, where each concept $c_k \in \mathcal{C}$ corresponds to a semantic category of interest (e.g., *red*, *vibrant*). For concept anchoring, we assume access to sparse supervision in the form of binary labels for a small subset of training samples: $\{(\boldsymbol{x}^{(i)}, \ell_{\mathcal{C}}^{(i)})\}$ where $\ell_{\mathcal{C}}^{(i)} \in \{0, 1\}^K$ indicates presence or absence of concepts at sample $\boldsymbol{x}^{(i)}$. Empirically, we find less than $0.1\%$ of samples need labels for any particular concept.

Consider an autoencoder architecture (figure 4 in section B.1) comprising encoder $f_\theta : \mathbb{R}^D \to \mathbb{R}^E$ and decoder $g_\phi : \mathbb{R}^E \to \mathbb{R}^D$, where $\theta$ and $\phi$ represent learned parameters. The forward pass operates as:

$$\boldsymbol{z} = f_\theta(\boldsymbol{x}), \quad \hat{\boldsymbol{z}} = \frac{\boldsymbol{z}}{\|\boldsymbol{z}\|_2}, \quad \hat{\boldsymbol{y}} = g_\phi(\hat{\boldsymbol{z}}) \tag{1}$$

where $\hat{\boldsymbol{z}} \in \mathbb{R}^E$ represents the normalized latent embedding constrained to the unit hypersphere $\mathbb{S}^{E-1}$. This normalization ensures consistent geometric properties throughout training and is known to work well in language models such as nGPT (Loshchilov et al., 2024).

The training objective integrates three components:

$$\mathcal{L}_{\text{total}}(\cdot) = \mathcal{L}_{\text{task}}(\cdot) + \mathcal{L}_{\text{structural}}(\hat{\boldsymbol{z}}) + \mathcal{L}_{\text{concept}}(\hat{\boldsymbol{z}}, \ell_{\mathcal{C}}) \tag{2}$$

where $\mathcal{L}_{\text{task}}$ maintains primary functionality (for our autoencoder: $\mathcal{L}_{\text{task}}(\boldsymbol{x}, \hat{\boldsymbol{y}}) = \|\boldsymbol{x} - \hat{\boldsymbol{y}}\|_2^2$), $\mathcal{L}_{\text{structural}}$ establishes global geometric properties, and $\mathcal{L}_{\text{concept}}$ positions specific concepts using sparse labels. We now detail each component.

## 2.2 STRUCTURAL CONSTRAINTS: ESTABLISHING GEOMETRIC FOUNDATION

Structural constraints provide a well-behaved geometric basis for latent representations through architectural design and loss-based regularization. Applied globally across all samples, these constraints create meaningful structure that supports concept organization and intervention.

**Unitarity Constraint** The explicit normalization in equation (1) projects all representations onto the unit hypersphere $\mathbb{S}^{E-1}$. This constraint, motivated by work showing that hypersphere representations improve separability and information preservation (Wang & Isola, 2020; Loshchilov et al., 2024), prevents representational collapse and establishes a space where cosine similarity measures semantic relationships—providing a natural framework for directional interventions. Unlike the regularization terms that follow, this architectural constraint ensures consistent geometric properties across all training phases.

**Separation Regularization** Without guidance, representations cluster in small regions of the hypersphere, yielding inseparable embeddings that resist intervention. We address this through a repulsion term:

$$\Omega_{\text{separate}}(\{\hat{\boldsymbol{z}}^{(i)}\}_{i=1}^{B}) = \frac{1}{B(B-1)} \sum_{i \neq j} (\hat{\boldsymbol{z}}^{(i)} \cdot \hat{\boldsymbol{z}}^{(j)})^p \tag{3}$$

where $i, j \in B$ are sample indices within a training batch. The exponent $p$ (set to 100 in our experiments) creates a sharp penalty discouraging high cosine similarity while minimally impacting orthogonal or weakly similar pairs, encouraging representations to spread across the hypersphere rather than clustering.

The complete structural loss is:

$$\mathcal{L}_{\text{structural}}(\hat{\boldsymbol{z}}^{(i)}) = \lambda_{\text{sep}} \, \Omega_{\text{separate}}(\{\hat{\boldsymbol{z}}^{(i)}\}) \tag{4}$$

where $\lambda_{\text{sep}}$ is a hyperparameter controlling the strength of the separation regularization.

## 2.3 Concept-Organizational Regularizers: Targeted Concept Positioning

Building on the structured foundation, we now position specific concepts within this space. These regularizers operate selectively on sparsely-labeled samples to create predictable concept locations enabling precise interventions.

**Anchor Regularization**    For concepts representable by a single prototype direction, we attract labeled examples toward fixed anchor points:

$$\Omega_{\text{anchor}}(\hat{\boldsymbol{z}}, \hat{\boldsymbol{v}}_c) = 1 - \hat{\boldsymbol{z}} \cdot \hat{\boldsymbol{v}}_c \tag{5}$$

where $\hat{\boldsymbol{v}}_c \in \mathbb{S}^{E-1}$ is the predetermined unit direction for concept $c \in \mathcal{C}$. This is minimized when $\hat{\boldsymbol{z}}$ aligns perfectly with the target direction.

**Subspace Regularization**    Some concepts are manifold and cannot be captured by a single direction (Engels et al., 2025). For these, we attract representations to predetermined axis-aligned subspaces:

$$\Omega_{\text{subspace}}(\hat{\boldsymbol{z}}, \mathcal{D}_c) = \sum_{i \notin \mathcal{D}_c} \hat{\boldsymbol{z}}_i^2 \tag{6}$$

where $\mathcal{D}_c \subset \{1, 2, \ldots, E\}$ specifies dimensions allocated to concept $c$. This supports subspaces of any dimensionality. Note that subspace constraints need not fully prescribe concept locations: confining concepts to dimensional subspaces reduces the search space for post-hoc methods while preserving flexibility within those dimensions.

**Repulsion Variants**    Both regularizers can be inverted to repel rather than attract:

$$\Omega_{\overline{\text{anchor}}}(\hat{\boldsymbol{z}}, \hat{\boldsymbol{v}}_c) = \max(0, \hat{\boldsymbol{z}} \cdot \hat{\boldsymbol{v}}_c) \tag{7}$$

$$\Omega_{\overline{\text{subspace}}}(\hat{\boldsymbol{z}}, \mathcal{D}_c) = \sum_{i \in \mathcal{D}_c} \hat{\boldsymbol{z}}_i^2 \tag{8}$$

The inverted anchor penalizes representations within 90° of the anchor direction, while inverted subspace pushes representations away from specific dimensions. In practice, attractive regularizers are applied to sparsely-labeled samples, while repulsive regularizers can be applied to all samples to reserve regions of latent space. Formally:

$$\ell_{c_k}^{(i)} = \begin{cases} 1 & \text{if } \Omega_{c_k} \text{ is repulsive} \\ 1 & \text{if } \Omega_{c_k} \text{ is attractive and } \boldsymbol{x}^{(i)} \in S_{c_k} \\ 0 & \text{otherwise} \end{cases} \tag{9}$$

where $S_{c_k}$ denotes the set of samples belonging to concept $c_k$, and $\Omega_{c_k}$ selects the appropriate concept regularizer based on the semantic properties of concept $c_k$.

The concept organizational loss aggregates regularizers across all concepts:

$$\mathcal{L}_{\text{concept}}(\hat{\boldsymbol{z}}^{(i)}, \ell_{\mathcal{C}}^{(i)}) = \sum_{k=1}^{K} \ell_{c_k}^{(i)} \lambda_{c_k} \Omega_{c_k}(\hat{\boldsymbol{z}}^{(i)}) \tag{10}$$

where $\ell_{c_k}^{(i)}$ controls which samples receive regularization for concept $c_k$, and $\lambda_{c_k}$ weights the strength of concept organization.

## 2.4 Complete Objective Function

Combining all components, the complete training objective becomes:

$$\mathcal{L}_{\text{total}}(\cdot) = \mathcal{L}_{\text{task}}(\cdot) + \lambda_{\text{sep}} \Omega_{\text{separate}}(\{\hat{\boldsymbol{z}}^{(i)}\}) + \sum_{k=1}^{K} \ell_{c_k}^{(i)} \lambda_{c_k} \Omega_{c_k}(\hat{\boldsymbol{z}}^{(i)}) \tag{11}$$

This formulation enables interpretable structure through minimal supervision: the vast majority of training samples ($> 99.9\%$ in our experiments) contribute only to task performance and global geometric structure, while a small fraction additionally guides concept organization without constraining representational flexibility for unlabeled data.

## 3 INTERVENTIONS

We demonstrate two intervention strategies that exploit the geometric structure established through Sparse Concept Anchoring. Because anchoring provides predetermined concept locations, these techniques operate without post-hoc analysis.

### 3.1 EXPERIMENTAL SETUP

We trained 60 color autoencoders per experiment with different random seeds, each using 4 or 5 latent dimensions ($E \in \{4, 5\}$). The *red* concept was anchored to the first dimension ($\hat{v}_{\text{red}} = (1, 0, ..., 0)$) using $83 \pm 8$ labeled examples per model (approximately $0.09\%$ of the dataset). To simulate realistic labeling conditions, we assigned labels stochastically during collation, allowing the same sample to receive different labels across batches.

Samples were drawn randomly with replacement from the full RGB cube throughout training, with selective regularization applied only to labeled samples. Hyperparameters (learning rate $\eta$, $\lambda_{\text{sep}}$, $\lambda_{c_k}$) varied according to tuned schedules detailed in section B.4.

*Optional enhancement.* In some experiments, we anchored *vibrant* colors to the subspace of the first two dimensions, labeling an additional $108 \pm 11$ samples ($\approx 0.11\%$). While not required for *red* interventions, this produced more interpretable latent structure by organizing hues as a color wheel. Results without *vibrant* anchoring appear in section C.5.1.

**Model Selection** For each experiment, we selected models from 60 training runs by identifying the Pareto frontier over intervention selectivity, validation reconstruction loss, and validation organization loss, then choosing the model with highest selectivity. Selectivity was measured as $R^2$ between reconstruction error and a power of similarity to *red*; details in sections C.3 and C.4.

**Evaluation Methodology** We evaluate interventions through reconstruction error (MSE) and visualization. For MSE, we compute:

$$\text{MSE}(\boldsymbol{x}, \boldsymbol{y}) = \frac{1}{3} \left[ (\boldsymbol{x}_r - \boldsymbol{y}_r)^2 + (\boldsymbol{x}_g - \boldsymbol{y}_g)^2 + (\boldsymbol{x}_b - \boldsymbol{y}_b)^2 \right] \tag{12}$$

For visualization, we use plots with a consistent three-panel layout throughout. Latent space projections (top) show axis-aligned views of the hypersphere; since our anchors are axis-aligned, these projections are directly interpretable without post-hoc rotation. Reconstruction grids (middle) display the model's output as each cell's background color, with a small inset square showing the true input color; where the two match, the model reconstructs faithfully. Error curves (bottom) plot MSE across hues at several brightness levels, so that spikes reveal which hues an intervention has disrupted.

Table 1: **Interventions selectively target *red*.** Reconstruction error (MSE) for two architectures: Anchored (section 3.2) uses attraction only; Isolated (section 3.3) adds repulsion to enable selective weight ablation. Both suppression and ablation increase error for *red* while preserving orthogonal colors.

| Color | | Baseline | Anchored (§3.2) | | Isolated (§3.3) | |
|---|---|---|---|---|---|---|
| | | | Suppression | Weight Ablation | Suppression | Weight Ablation |
| Red | 🟥 | 0.00 | +0.28 | +0.33 | +0.21 | +0.34 |
| Lime | 🟩 | 0.00 | +0.00 | +0.00 | +0.00 | +0.00 |
| Cyan | 🟦 | 0.00 | +0.00 | +0.34 | +0.00 | +0.00 |
| Purple | 🟪 | 0.00 | +0.00 | +0.00 | +0.00 | +0.00 |
| Black | ⬛ | 0.00 | +0.00 | +0.00 | +0.00 | +0.00 |
| Gray | ⬛ | 0.00 | +0.00 | +0.00 | +0.00 | +0.00 |
| White | ⬜ | 0.00 | +0.00 | +0.00 | +0.00 | +0.00 |

### 3.2 ANCHORED ARCHITECTURE

The anchored architecture uses attraction regularizers only, drawing labeled samples toward their target directions. We evaluate two intervention types: *suppression*, which modifies activations at inference, and *weight ablation*, which permanently zeros weights for anchored dimensions.

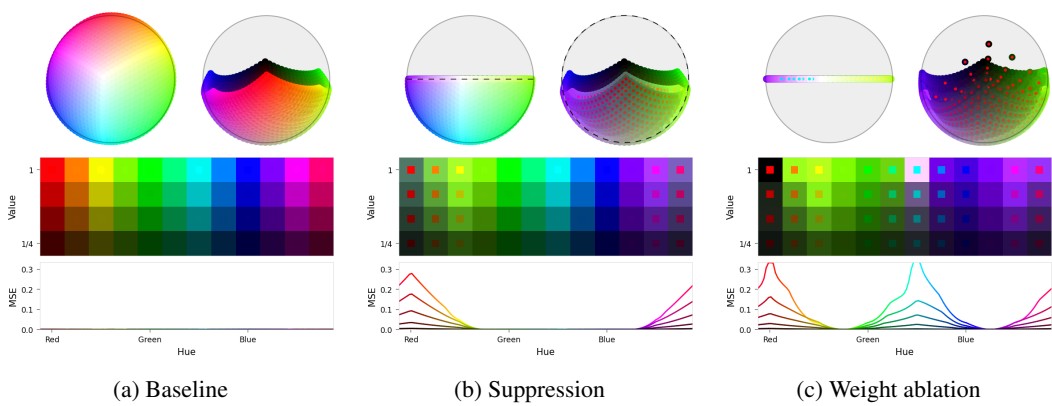

(a) Baseline       (b) Suppression       (c) Weight ablation

Figure 2: **Concept interventions in structured latent space.** A 4-dimensional autoencoder with *red* and *vibrant* anchored. **(a)** Baseline: organized color wheel with near-zero reconstruction error. **(b)** Suppression: *red* hues reconstruct as dark gray (inset squares retain the true color for comparison), producing the error spike near *red*. **(c)** Weight ablation affects both *red* and *cyan*; additional constraints enable selective deletion (section 3.3).

**Suppression** Suppression modifies latent activations during the forward pass without changing model weights.[2] We project out the component of a latent activation aligned with a target concept direction. For unit-normalized latent activations $\hat{z} \in \mathbb{R}^E$ and concept vector $\hat{v} \in \mathbb{R}^E$ with $\|\hat{z}\|_2 = \|\hat{v}\|_2 = 1$:

$$\hat{z}' = \hat{z} - (\hat{z} \cdot \hat{v})\hat{v} \tag{13}$$

This removes the component aligned with $\hat{v}$ while preserving orthogonal information.

Geometrically, suppression pushes activations off the hypersphere, placing them off-manifold. This contrasts with weight ablation (section 3.3), which maintains on-manifold geometry through renormalization. Since off-manifold activations force the decoder to rely on its biases, and assuming these produce *middle gray* $(0.5, 0.5, 0.5)$, we expect reconstruction error for *red* around $\frac{1}{4}$:

$$\text{MSE}(\boldsymbol{x}_{\text{red}}, (0.5, 0.5, 0.5)) = \frac{1}{3}\left[(1-0.5)^2 + (0-0.5)^2 + (0-0.5)^2\right] = \frac{1}{4} \tag{14}$$

**Model Architecture** The encoder and decoder each had one hidden layer with 16 units. Latent space was four-dimensional ($E = 4$), with *red* anchored at $\hat{v}_{\text{red}} = (1, 0, 0, 0)$ and *vibrant* colors constrained to dimensions $\mathcal{D}_{\text{vibrant}} = \{1, 2\}$. All 60 training runs induced the target geometry.

Suppression was consistently selective ($R^2 = 0.95 \pm 0.02$; figure 11); see sections C.3 and C.4. Pareto frontier analysis identified 4 non-dominated models, from which we selected the one with highest intervention selectivity.

**Results** Suppression achieves highly selective concept attenuation (figure 2 and table 1). Reconstruction error for *red* increases from 0.000 to 0.284, approaching the theoretical bound of 0.25 and substantially disrupting red appearance. Orthogonal colors remain unaffected: *lime* and *purple* maintain MSE of 0.000. Error correlates strongly with squared color similarity to *red* ($R^2 = 0.99$), confirming the quadratic relationship expected from the projection mechanism (see section C.3).

In this model, weight ablation increases error for both *red* and *cyan*, demonstrating that additional constraints are needed for selective concept deletion—which we address in the next section.

### 3.3 ISOLATED ARCHITECTURE

Unlike suppression, weight ablation requires exclusive use of the target dimension—other concepts can leak into it and be disrupted when its weights are zeroed (section 3.2). The isolated architecture addresses this by adding repulsion regularizers that push all samples away from the anchored dimension, reserving it exclusively for the target concept and enabling selective weight ablation.

---

[2]We also evaluate repulsion, another behavioral steering technique, in section C.4.

**Weight Ablation**  Weight ablation permanently removes concepts by zeroing weights that produce and consume activations in targeted latent dimensions.[3] For target dimensions $\mathcal{D}_c$ anchoring concept $c$:

$$\mathbf{W}_f[d,:] = \mathbf{0}, \quad b_f[d] = 0, \quad \mathbf{W}_g[:,d] = \mathbf{0} \quad \forall d \in \mathcal{D}_c \tag{15}$$

where $\mathbf{W}_f$ and $b_f$ are the encoder output weights and bias, and $\mathbf{W}_g$ is the decoder input weights. Unlike suppression, weight ablation maintains on-manifold geometry through renormalization.

Weight ablation doesn't redirect activations to any particular alternative concept. Assuming the resulting direction is random, the expected reconstruction error should be $\frac{1}{3}$:

$$\mathbb{E}[\text{MSE}(\boldsymbol{x}_{\text{red}}, \boldsymbol{y}_{\text{random}})] = \frac{1}{3}\mathbb{E}\left[(1-r)^2 + g^2 + b^2\right] = \frac{1}{3} \tag{16}$$

where $(r, g, b) \sim \text{Uniform}[0,1]^3$.

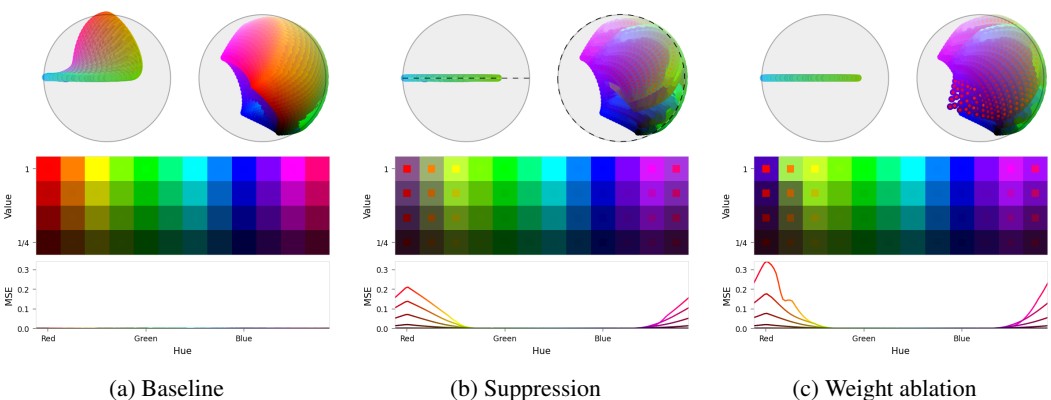

(a) Baseline    (b) Suppression    (c) Weight ablation

Figure 3: **Isolated architecture enables selective permanent removal.** A 5-dimensional autoencoder with *red* anchored and repulsive terms applied. Same layout as figure 2. **(a)** Successful concept organization. **(b)** Suppression eliminates *red*. **(c)** Weight ablation eliminates *red* by zeroing its dimension. Both interventions are highly selective.

**Model Architecture**  The encoder and decoder each had two hidden layers with 10 units. Latent space was five-dimensional ($E = 5$), with *red* anchored at $\hat{\boldsymbol{v}}_{\text{red}} = (1, 0, 0, 0, 0)$. To isolate *red*, we applied anchor attraction plus two repulsion terms pushing all samples away from the *red* dimension and direction. Early training emphasized repulsion to clear the target dimension, then shifted to anchor attraction (see section B.4). All 60 training runs induced the target geometry, though with greater variability than the anchored architecture.

Ablation selectivity also varied more than suppression ($R^2 = 0.86 \pm 0.15$; figure 13); see sections C.3 and C.4. Pareto frontier analysis identified 22 non-dominated models, from which we selected the one with highest intervention selectivity.

**Results**  Weight ablation achieves complete elimination of *red* while preserving orthogonal colors (figure 3 and table 1). Reconstruction error for *red* increases from 0.000 to 0.343, approaching the theoretical bound of $0.\overline{3}$ for random on-manifold directions. Orthogonal and opposing colors—including *cyan*, which opposes *red* in both RGB and HSV spaces—show no measurable degradation. Error exhibits a strong cubic relationship with color similarity to *red* ($R^2 = 0.98$), confirming selective concept removal (see section C.3).

### 3.4  INTERVENTION TRADE-OFFS

The two architectures address complementary needs. Attraction regularizers alone (section 3.2) suffice for reversible, inference-time suppression. Permanent removal via weight ablation requires the additional repulsive regularizers of the isolated architecture (section 3.3) to reserve the target dimension exclusively, at the cost of additional training complexity. The choice between the two therefore depends on whether the application requires dynamic steering or permanent removal.

---

[3]We also evaluate pruning, which removes dimensions entirely, in section C.2.

## 4 DISCUSSION

The predictable geometry established through Sparse Concept Anchoring opens new paths for model control. Traditional interpretability workflows identify concepts post-hoc, then design interventions around discovered structure. By anchoring concepts during training, we invert this relationship: structure enables intervention by design.

**Variability and Fallback Control**   The two intervention modes serve complementary roles (section 3.4), but weight ablation selectivity varies more (figure 13) because renormalization redistributes removed components. Both would benefit from explicit fallback control: suppression relies on decoder biases (limiting generality), while ablation's redistribution is unreliable. Optimal ablation (Li & Janson, 2024) could address this by replacing removed components with optimized constants.

### 4.1 LIMITATIONS AND FUTURE WORK

**Permanence**   While ablation permanently removes information, it may leave structural traces aiding adversarial recovery. SCA's clean geometry might paradoxically increase this risk: examining encoder activations could reveal smaller pre-normalization values for certain inputs, exposing removed capabilities. Whether this enables practical attacks remains open.

**Extension to Other Domains**   Our experiments benefit from RGB data, where concept similarity and labels are unambiguous. Linguistic domains introduce higher-dimensional manifolds, context-dependent similarity, and subjective labeling—the fraction of labeled examples required for anchoring may increase with manifold dimensionality and curvature. Applying SCA to transformers requires consideration of where to add geometric constraints—for example, at which layers of the residual stream—and whether anchoring can resist bypass via residual connections. Our initial approach might gather concept labels for passages using source metadata or automated labeling, then apply weak regularization to all tokens in labeled passages. Given SCA's robustness to label noise this may prove sufficient.

Longer-term validation should target GPT-2–scale transformers and progressively more abstract concepts. As the number of anchored concepts $K$ grows, the cumulative effect of competing attraction and repulsion terms may create a difficult optimization landscape; characterizing both data-requirement scaling and the point at which anchoring degrades task performance are important open questions. Multi-dimensional concepts can anchor subspaces (as with *vibrant* colors), which may mitigate $K$-scaling by reducing the effective number of competing terms, while fuzzy concepts might benefit from weak subspace anchoring that draws related samples to known regions to reduce the post-hoc search space.

## 5 CONCLUSION

Sparse Concept Anchoring inverts traditional interpretability: rather than discovering concept locations post-hoc, it establishes predictable geometric structure during training through minimal supervision. By fixing concept directions and regularizing representations, SCA enables direct interventions on model behavior.

Experiments on color autoencoders show that both inference-time suppression and parameter-level ablation achieve selective concept modification with minimal collateral impact. Sparse, noisy labels ($< 0.1\%$ of data) shaped latent geometry into actionable subspaces, demonstrating that anchoring few concepts suffices for controllable representations while preserving flexibility.

Although current experiments use low-dimensional settings, geometric regularization with sparse supervision provides a path toward scalable concept-level control. For safety-critical applications requiring auditable, modifiable systems, proactively shaping representations may prove more tractable than reactive analysis of emergent structure.

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

## A   RELATED WORK

Our work sits at the intersection of several active research areas: methods for building interpretability into models during training, techniques for steering model behavior through representation manipulation, and approaches for removing specific model capabilities.

### A.1   CONCEPT-BASED INTERPRETABILITY METHODS

Several families of methods aim to make neural networks interpretable through human-meaningful concepts. Concept Bottleneck Models (Koh et al., 2020) enforce interpretability architecturally by introducing an intermediate layer where each dimension corresponds to a predefined concept, enabling test-time interventions—though originally requiring full supervision, recent work has reduced this burden through post-hoc discovery or sparse training-time methods with minimal labels (Oikarinen et al., 2023; Semenov et al., 2024; Sawada & Nakamura, 2022). However, post-hoc interpretability methods have revealed that concepts in CBMs may not correspond to semantically meaningful input features (Margeloiu et al., 2021), motivating approaches that explicitly structure concept representations during training. Concept Activation Vectors (Kim et al., 2018) take a lightweight post-hoc approach, learning linear probes from as few as 30 examples per concept to identify where concepts appear in trained models—useful for bias detection but providing no architectural guarantees for interventions. Sparse Autoencoders use unsupervised dictionary learning to discover interpretable features models actually use, recently scaling to frontier language models (Huben et al., 2024; Templeton et al., 2024), though features are discovered rather than positioned during training. Concept Whitening (Chen et al., 2020) replaces batch normalization with transformations that align latent space axes with concepts using representative examples, enabling layer-wise interpretability without hurting performance. These methods trade off supervision requirements, timing of concept incorporation (training vs. post-hoc), and intervention capabilities.

### A.2   MACHINE UNLEARNING AND REPRESENTATION ENGINEERING

Machine unlearning addresses removing specific capabilities from trained models, driven by privacy regulations and safety concerns. Gradient-based methods attempt to reverse training through gradient ascent on "forget" data (Jang et al., 2022; Yao et al., 2024; Zhang et al., 2024), but face challenges with gradient explosion, catastrophic forgetting, and instability—particularly at high forget rates. Representation-based methods like RMU (Li et al., 2024) redirect activations of unwanted content toward random directions, reducing hazardous knowledge to near-random performance on benchmarks like WMDP, though the distinction between masking and true removal remains unclear. Training-time approaches remain rare: SISA (Bourtoule et al., 2021) enables efficient removal through data sharding but with substantial computational overhead, while Ready2Unlearn (Duan et al., 2025) uses meta-learning to prepare models for later unlearning—yet both operate through data organization or optimization dynamics rather than explicit geometric positioning. Representation engineering methods manipulate behavior by modifying internal activations (Zou et al., 2023): activation addition (Turner et al., 2023) extracts steering vectors from paired prompts, while optimized methods like BiPO (Cao et al., 2024) and reinforcement learning approaches (Sinii et al., 2025) train better steering vectors—but all depend on directions discovered in already-trained models. Systematic analysis reveals substantial reliability issues: steering effectiveness varies dramatically across inputs, many concepts prove "anti-steerable", and success often depends on spurious correlations rather than coherent concepts (Tan et al., 2024). Abliteration (Arditi et al., 2024) demonstrated that safety behaviors can be removed through targeted weight orthogonalization with negligible performance degradation, providing evidence for the linear representation hypothesis—yet achieving selective ablation without side effects remains challenging when features are distributed or when networks exhibit "compensatory masquerade" by routing around ablations (Meyes et al., 2019).

### A.3   GEOMETRIC CONSTRAINTS IN NEURAL NETWORKS

Recent work has explored how geometric constraints on representations improve training dynamics and enable interpretability. nGPT (Loshchilov et al., 2024) normalizes all transformer components to unit norm, constraining representations to a hypersphere, yielding 4-20× faster convergence, more interpretable angular relationships, and stable gradients—suggesting hypersphere constraints improve

both interpretability and optimization itself. Angular margin losses from face recognition (Liu et al., 2017; Deng et al., 2022) enforce separation between classes in hyperspherical geometry through L2-normalized features and additive margins, achieving state-of-the-art results because angular constraints create geometrically clean separation. Theoretical analysis shows contrastive learning on hyperspheres naturally optimizes for alignment and uniformity (Wang & Isola, 2020)—properties that facilitate linear separability and robust representations. In the context of concept-based models, concept orthogonal loss (Sheth & Kahou, 2023) encourages separation between learned concept representations while reducing intra-concept distance, improving concept disentanglement in CBMs through auxiliary training objectives—though applied to dense concept bottlenecks rather than sparse, pre-positioned concepts. While geometric constraints have improved training efficiency and discriminability, their use specifically for positioning concepts to enable interventions—particularly with minimal supervision—remains less explored.

### A.4    Positioning Our Work

Our method integrates training-time geometric positioning of concepts through hypersphere constraints with sparse supervision ($< 0.1\%$ of examples labeled per concept). We build most directly on sparse CBM work that reduces supervision requirements (Sparse-CBM, UCBM, Z-CBM) and on geometric representation learning (nGPT, angular margin methods) that uses hypersphere constraints for improved training and interpretability. The closest related work differs in key trade-offs: CBMs and variants provide training-time structure but vary in supervision needs and intervention capabilities; Concept Whitening uses whitening transformations rather than hypersphere normalization and requires moderate supervision through concept datasets; representation engineering methods discover steering directions post-hoc, inheriting reliability issues from emergent geometry; training-time unlearning methods (Ready2Unlearn, SISA) use meta-learning or data organization rather than concept positioning.

Our observation-driven approach to hyperparameter tuning—watching latent geometry evolve during training to manually adjust time-varying regularizer schedules—represents a form of developmental interpretability. This differs from recent work studying how data distribution changes during training affect final models; we instead manipulate optimization dynamics through coordinated loss term schedules while training on fixed data. The motivation for training-time intervention comes from evidence that neural networks lose plasticity early in training and establish connectivity patterns during critical periods (Achille et al., 2019), making post-hoc modification difficult.

Our method addresses three specific gaps. First, unlike sparse CBMs that discover concepts during training, we *fix* concept locations a priori, enabling pre-planned interventions. Second, unlike CAVs and steering vectors that depend on emergent geometry, we *construct* geometry that supports interventions by design. Third, unlike machine unlearning methods that attempt post-hoc removal, we achieve selective ablation by isolating concepts during training—preventing the entanglement that makes clean removal difficult in standard architectures. Whether this approach scales beyond color autoencoders to complex domains like language models—particularly whether geometric constraints remain tractable in high-dimensional spaces with attention and residual pathways—remains an important open question for future work.

## B    Training Details

### B.1    Visualization of the architecture

Figure 4 illustrates the architecture of the autoencoder used in our experiments.

The architecture takes 3D RGB inputs $x$ through an encoder $f_\theta$ with 1-2 fully-connected hidden layers with GeLU activations, projecting to 4-5D latent activations $z$. An L2 normalization layer $\circledN$ constrains these to the unit hypersphere, yielding $\hat{z}$. The decoder $g_\phi$ mirrors this structure, mapping from the normalized latent space back to RGB through 1-2 hidden layers with GeLU activations and a final linear projection to outputs $\hat{y}$. Outputs are unconstrained during training but clamped to $[0, 1]$ per channel during evaluation.

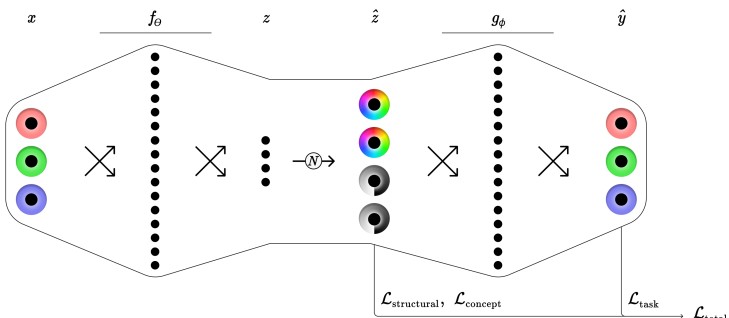

Figure 4: **Spherical Autoencoder Architecture.** The encoder maps RGB inputs through hidden fully-connected layers ($\times$) to 4D activations, which are explicitly normalized ($N$) to constrain latent representations to the unit hypersphere. The decoder reconstructs RGB outputs from these normalized latent representations.

## B.2   VISUALIZATION OF THE STRUCTURAL CONSTRAINTS AND ORGANIZATIONAL REGULARIZERS

Figures 5 and 6 illustrate the geometric effects of the regularization terms introduced in section 2. Structural biases (figure 5) show how unitarity constrains embeddings to the hypersphere while separation applies repulsive forces to prevent clustering. Organizational biases (figure 6) demonstrate the directional effects: anchor and subspace terms create attractive forces toward specific points and planes respectively, while their anti-variants apply repulsive forces away from designated regions.

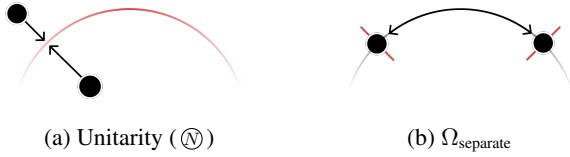

(a) Unitarity ($N$)                    (b) $\Omega_{\text{separate}}$

Figure 5: **Structural Biases.** *a*: Unitarity places embeddings ($\bullet$) on the unit hypersphere ($\bigcirc$). *b*: Separation repels pairs of embeddings from each other to reduce clustering.

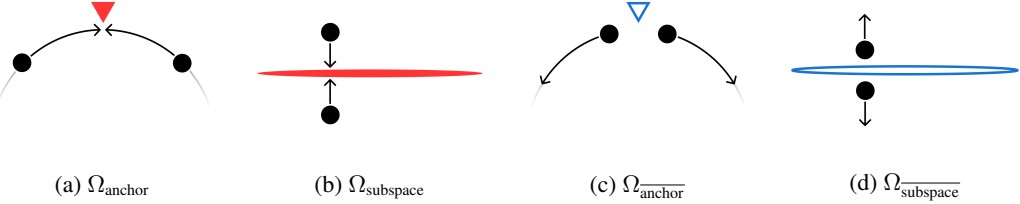

(a) $\Omega_{\text{anchor}}$          (b) $\Omega_{\text{subspace}}$          (c) $\Omega_{\overline{\text{anchor}}}$          (d) $\Omega_{\overline{\text{subspace}}}$

Figure 6: **Organizationl Biases**. *a*: Anchor applies rotational attraction of embeddings ($\bullet$) to a fixed point on the hypersphere ($\blacktriangledown$). *b*: Subspace applies linear attraction to a set of embedding dimensions ($\rule{12pt}{3pt}$). *c*: Anti-anchor applies rotational repulsion from a fixed point on the hypersphere ($\triangledown$). *d*: Anti-subspace applies linear repulsion from a set of embedding dimensions ($\ominus$). All are regularization loss terms.

## B.3   DETAILS ON MINIMAL SUPERVISION

Our framework applies organizational biases selectively using sparse, stochastic supervision to simulate realistic deployment scenarios where comprehensive concept labeling is infeasible. This approach demonstrates that meaningful latent structure can emerge from weak supervision signals, addressing a critical bottleneck in scaling interpretability methods to large models.

**Continuous Label Probability Design**  We design concept-specific probability functions that capture the underlying geometric properties of our target concepts. For the *red* concept, we define:

$$p_{\text{red}}(\boldsymbol{x}) = 0.08 \left[ \boldsymbol{x}_r \left( 1 - \frac{\boldsymbol{x}_g}{2} - \frac{\boldsymbol{x}_b}{2} \right) \right]^8 \tag{17}$$

where $\boldsymbol{x}_r, \boldsymbol{x}_g, \boldsymbol{x}_b \in [0, 1]$ are normalized RGB components. This formula achieves maximum probability for pure red ($\boldsymbol{x}_r = 1, \boldsymbol{x}_g = 0, \boldsymbol{x}_b = 0$) with rapid exponential decay as other color components increase, creating a sharp but imperfect concept boundary.

For experiments requiring broader hue organization, we define a complementary *vibrant* concept:

$$p_{\text{vibrant}}(\boldsymbol{x}) = 0.01(\boldsymbol{x}_s \ \boldsymbol{x}_v)^{10} \tag{18}$$

where $\boldsymbol{x}_s$ and $\boldsymbol{x}_v$ represent saturation and value in HSV color space. The extreme exponent creates a sharp distinction between fully saturated colors and even slightly desaturated ones, enabling precise control over which samples receive vibrant labels.

**Stochastic Label Generation**  These continuous probabilities are stochastically discretized during training to create realistic weak supervision. For each sample and concept, we generate binary labels through probabilistic sampling:

$$\ell_c(\boldsymbol{x}) = \mathbf{1}[p_c(\boldsymbol{x}) > u], \quad u \sim \text{Uniform}(0, 1) \tag{19}$$

This process creates inherently sparse supervision: pure red receives the *red* label $\ell_{\text{red}}$ only $8\%$ of the time on average, while colors progressively distant from red are labeled with rapidly decreasing frequency. The resulting supervision covers only $0.08\%$ of training samples for our primary experiments, yet proves sufficient to induce global latent organization.

**Realistic Supervision Simulation**  This stochastic approach deliberately simulates real-world labeling scenarios encountered in large-scale model training, such as automated sentiment analysis of web text or crowd-sourced annotation with inherent inconsistencies. Our method's robustness to weak, noisy supervision suggests practical applicability to domains where perfect concept labels are unavailable or prohibitively expensive to obtain. Our experiments demonstrate that minimal signals can effectively structure latent representations, providing a pathway toward intrinsic interpretability in production systems.

### B.4 Optimizer Configuration and Hyperparameter Schedules

Training employed the Adam optimizer with the default PyTorch options: $\beta_1 = 0.9$, $\beta_2 = 0.999$, $\epsilon = 10^{-8}$, and $\lambda_{wd} = 0$ (no weight decay). We used a batch size of 64, which directly impacts $\Omega_{\text{separate}}$ since it computes pairwise interactions within each batch. All experiments ran for 1,500 training steps.

**Time-Varying Loss Terms**  Both the learning rate $\eta$ and regularizer weights ($\lambda_{\text{sep}}$, $\lambda_{\text{anchor}}$, $\lambda_{\text{subspace}}$, $\lambda_{\overline{\text{anchor}}}$, $\lambda_{\overline{\text{subspace}}}$) varied according to coordinated schedules throughout training. We specified these schedules as transition timelines (inspired by animation keyframe systems), allowing synchronized changes across multiple hyperparameters and straightforward experimental iteration.

The learning rate followed a regular pattern for all experiments: brief warmup from $10^{-8}$ to 0.01 over the first 10 steps, ramp to 0.1 until step $\sim$750, maintain through the main training phase, then decay to 0.05 by step 1500.

Regularizer weights had coordinated but distinct trajectories. In experiments requiring dimensional clearing (e.g., the 1D weight ablation experiment in section 3.3), the anti-subspace term was initially strong to reserve the target dimension, then reduced to near-zero at step 750; with the anchor term becoming dominant around step 200 (see figure 7b).

In simpler experiments without explicit clearing requirements (e.g., the suppression experiment in section 3.2), regularizer weights still varied substantially: the structural separation weight was held low and steady at first but decayed to near-zero over the second half of training; while organizational term weights were strongest mid-training to establish concept positions before relaxing to allow fine-tuning on the primary objective (see figure 7a).

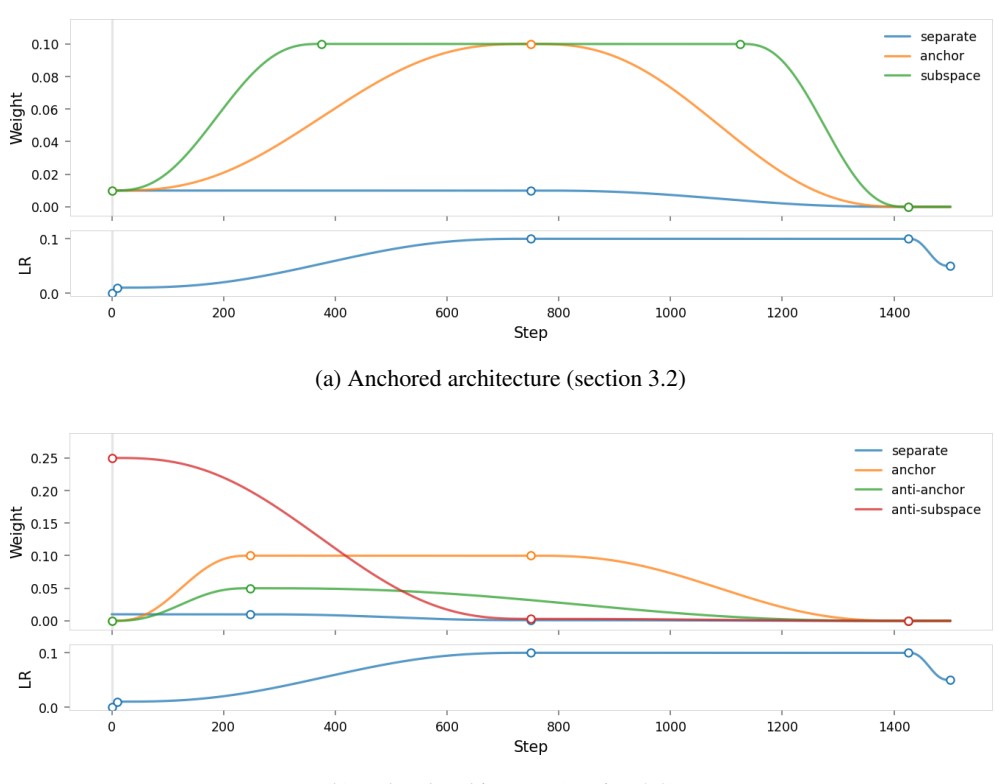

(a) Anchored architecture (section 3.2)

(b) Isolated architecture (section 3.3)

Figure 7: **Managing multiple loss terms with varying weights.** We emphasized different regularizers at different phases of model development. **(a)** A consistently high subspace weight encouraged formation of the color wheel; anchor weight peaked mid-training to rotate it to align *red* with the target direction. **(b)** A high initial anti-subspace weight reserves target dimensions for concept anchoring; later, the anchor weight dominates to pull concept representations into position.

These time-varying schedules proved *necessary for all experiments*: We were unable to find good fixed-weight configurations. The improvement from varying hyperparameters suggests that the changing loss landscape helps the optimizer to navigate competing objectives between task performance, structural constraints, and concept positioning. Our empirical observation of training dynamics—including visualization of evolving latent geometry—enabled effective manual tuning. This approach to hyperparameter configuration through observation of training behavior represents a form of developmental interpretability that may complement theoretical frameworks for understanding neural network training.

Other hyperparameters are detailed elsewhere: the separation exponent $p = 100$ in section 2, and the label generation process in section B.3.

## B.5 TRAINING DATA

Training data were drawn from the RGB color cube, which encompasses all colors representable in the RGB color model. Each sample is represented as a 3D vector in RGB space, with each channel in $[0, 1]$. The cube was subdivided uniformly along each axis, and training samples were generated at the grid points. In our experiments, we used a $8 \times 8 \times 8$ grid, yielding $512$ training samples, shown in figure 8.

Superficially, the cube resembles some of our latent space plots—however the spaces are quite different: the RGB cube is solid (the magnitudes of $x$ and $y$ are significant), whereas our latent spaces are hyperspherical ($\hat{z}$ is purely directional).

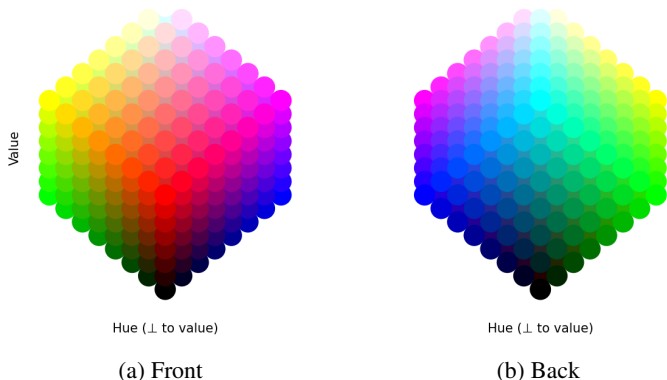

Value

Hue (⊥ to value)                    Hue (⊥ to value)

(a) Front                          (b) Back

Figure 8: **The RGB cube as training data.** Two views of the cube are shown, both oriented such that the black-to-white diagonal runs from bottom-to-top; thus red, blue, and green are nearer the bottom, whereas cyan, yellow, and magenta are nearer the top. Grays are located in the center of the cube (not visible). *a*: View facing the warm hues, with red in the middle and yellow and magenta on either side. *b*: View facing the cool hues, with cyan in the middle and blue and green on either side.

## C  INTERVENTION DETAILS

### C.1  BEHAVIORAL STEERING: DETAILED FORMULATIONS

This appendix provides detailed mathematical formulations and geometric visualizations for the behavioral steering interventions introduced in section 3.2. While the main text presents simplified versions of suppression and mentions repulsion conceptually, here we present the full parametric families of intervention functions and their geometric properties.

**Suppression.**  For latent activations $\hat{z} \in \mathbb{R}^E$ and concept vector $\hat{v} \in \mathbb{R}^E$ with $\|\hat{z}\|_2 = \|\hat{v}\|_2 = 1$, the general suppression transformation is:

$$\hat{z}' = \hat{z} - h(\alpha)(\hat{z} \cdot \hat{v})\hat{v} \tag{20}$$

where $\alpha = \max(0, \hat{z} \cdot \hat{v})$ represents the alignment (positive cosine similarity) between the activation and concept direction, and $h(\alpha)$ is a suppression strength function that maps alignment to intervention intensity.

The main text uses the aggressive formulation $h(\alpha) = \alpha$, which completely nullifies the aligned component. However, we can implement bounded falloff functions that create smooth transitions from no suppression to maximum intervention:

$$h(\alpha) = \begin{cases} 0 & \text{if } \alpha < a \\ b\left(\frac{\alpha-a}{1-a}\right)^p & \text{if } \alpha \geq a \end{cases} \tag{21}$$

where $a \in [0, 1)$ is the alignment threshold below which no suppression occurs, $b \in [0, 1]$ is the maximum suppression strength, and $p \geq 0$ controls the falloff curve shape.

This design preserves representations with low concept alignment while progressively suppressing those with higher alignment. The geometric effect is that suppressed activations are pushed off the unit hypersphere, placing them off-manifold relative to the distribution learned during training (figure 9).

**Repulsion.**  Unlike suppression, which pushes embeddings off the hypersphere, repulsion rotates activations away from concept directions to new positions while preserving unit norm:

$$\hat{z}' = m(\alpha)\hat{v} + \sqrt{1 - m(\alpha)^2}\hat{\mathbf{u}}_\perp \tag{22}$$

where $m(\alpha)$ is a mapping function that determines the target alignment, and $\hat{\mathbf{u}}_\perp$ is the unit vector perpendicular to $\hat{v}$ in the plane spanned by $\hat{z}$ and $\hat{v}$:

$$\hat{\mathbf{u}}_\perp = \frac{\hat{z} - (\hat{z} \cdot \hat{v})\hat{v}}{\|\hat{z} - (\hat{z} \cdot \hat{v})\hat{v}\|} \tag{23}$$

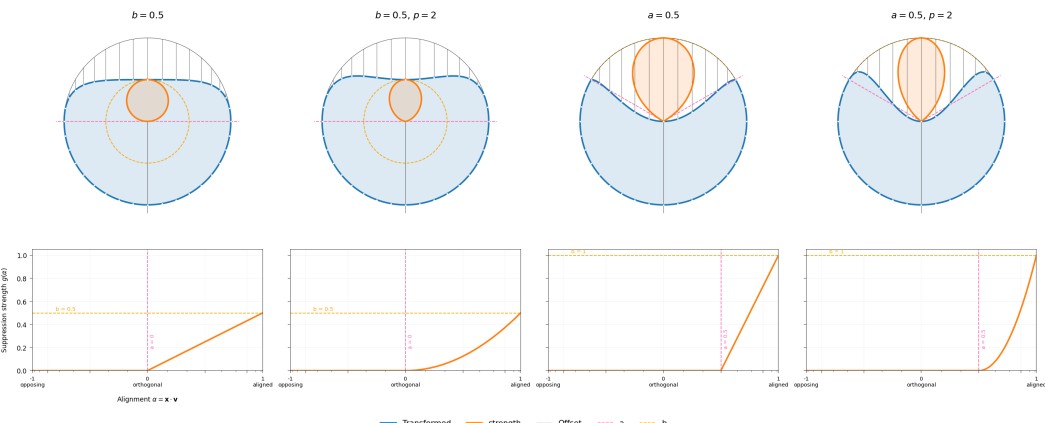

Figure 9: **Suppression Intervention Lobes.** *Top*: Polar projections where the angular coordinate represents the direction of a unit input vector, and the radial coordinate shows magnitude. The orange region shows suppression strength, while the blue region shows transformed activations, with straight lines showing the transformation from original to suppressed activations (white dots). *Bottom*: Suppression strength as a function of alignment.

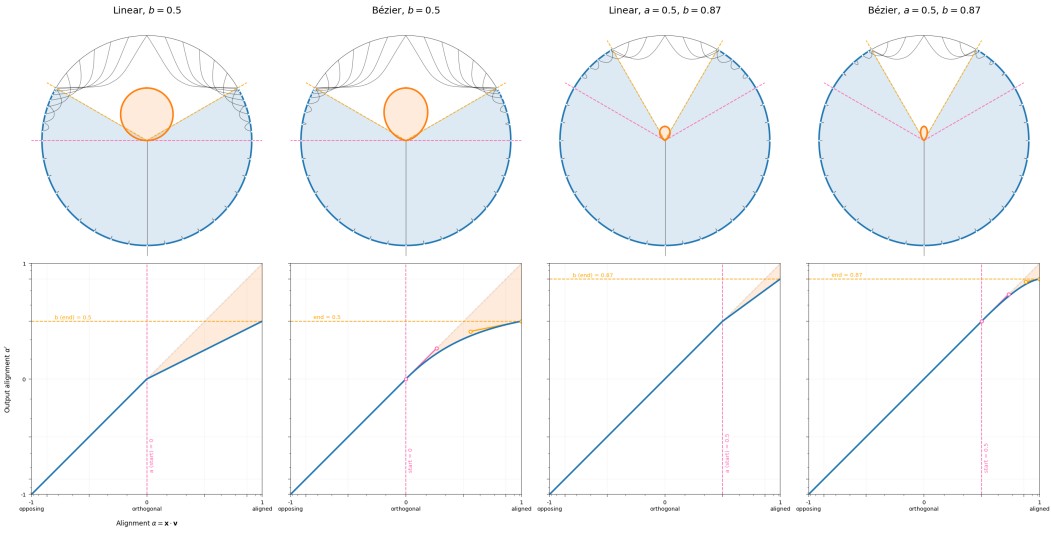

Figure 10: **Repulsion Intervention Lobes.** *Top*: Polar plots show how vectors are rotated to new positions on the unit hypersphere, with curved "chord" lines illustrating the rotation paths from input to output positions (white dots). *Bottom*: Mapping functions $m(\alpha)$ that determine target alignments. The columns alternate between using linear mappers and Bézier mappers. The filled regions between the identity line and mapping curve indicate the magnitude of alignment reduction.

Activations are rotated within the 2D plane defined by the original activation and the concept vector, moving them away from the concept direction while staying on the unit hypersphere.

The mapping function $m(\alpha)$ provides flexible control over post-intervention alignment. We implement linear mappings that repel activations from high-alignment regions:

$$m_{\text{linear}}(\alpha) = \begin{cases} \alpha & \text{if } \alpha < a \\ b & \text{if } \alpha \geq a \end{cases} \tag{24}$$

where $a$ is the threshold and $b \geq 0$ is the maximum mapped value, creating a "ceiling effect" that prevents excessive alignment with the concept vector.

Alternatively, we can use cubic Bézier curves for smoother transitions. The Bézier mapping function is defined as:

$$m_{\text{Bézier}}(\alpha) = \begin{cases} \alpha & \text{if } \alpha \leq a \\ B_y(t^*) & \text{if } \alpha > a \end{cases} \tag{25}$$

where the curve $B(t) = (B_x(t), B_y(t))$ is parameterized by control points $P_0 = (a, a)$, $P_3 = (1, b)$, and interior points $P_1, P_2$ chosen to satisfy desired endpoint slopes. The standard cubic Bézier formulation is:

$$B(t) = (1-t)^3 P_0 + 3(1-t)^2 t P_1 + 3(1-t)t^2 P_2 + t^3 P_3, \quad t \in [0, 1] \tag{26}$$

For a given $\alpha > a$, we find $t^* \in [0, 1]$ such that $B_x(t^*) = \alpha$, and the mapped value is $m_{\text{Bézier}}(\alpha) = B_y(t^*)$. By setting control points appropriately (e.g., with unit start slope and flat end slope), we obtain smooth, monotone mappings that gradually reduce alignment. For instance, choosing $a = \cos(90°) = 0$ and $b = \cos(60°) = 0.5$ yields a smooth map from $[\cos(90°), \cos(0°)] = [0, 1]$ to $[\cos(90°), \cos(60°)] = [0, 0.5]$ with desirable geometric properties.

Repulsion creates a radius $\sin^{-1}(b)$ "hole" in latent space around the concept direction, redirecting activations to nearby regions while preserving the manifold structure and resulting in a high-density ring between alignments $a$ and $b$ (figure 10).

## C.2 PERMANENT CONCEPT REMOVAL: IMPLEMENTATION DETAILS

This appendix provides additional implementation details for the permanent concept removal techniques presented in section 3.3.

Weight ablation and pruning both eliminate concepts by severing the information pathways through targeted latent dimensions. For target dimension $d$ anchoring a specific concept, we must prevent both the *production* of activations in dimension $d$ by the encoder and the *consumption* of information from dimension $d$ by the decoder.

**Ablation.** Ablation achieves this by zeroing the relevant weight matrix entries and bias terms (see equation (15) in section 3.3). This maintains the architectural structure and dimensionality of all intermediate computations.

**Pruning.** Pruning operates by wholly removing the targeted dimension from the architecture. For target dimensions $\mathcal{D} \subset \{0, 1, \ldots, E - 1\}$, pruning constructs reduced weight matrices $\mathbf{W}'_{\text{enc}} \in \mathbb{R}^{(E-|\mathcal{D}|) \times H}$ and $\mathbf{W}'_{\text{dec}} \in \mathbb{R}^{H \times (E-|\mathcal{D}|)}$ containing only the rows and columns corresponding to non-targeted dimensions. This reduces the model's latent dimensionality from $E$ to $E - |\mathcal{D}|$.

Both methods eliminate the targeted concept's contribution to model outputs, and are functionally equivalent. Ablation is somewhat simpler to implement, but pruning may be preferable if one wishes to hide the existence of the removed concept—although in that case, we expect the removed concept to have left a lasting imprint on the representations of the remaining dimensions, which may still be detectable.

## C.3 COLOR SIMILARITY METRIC

To validate that our method preserves logical structure, we measure the relationship between the similarity of inputs to the target concept and reconstruction error under intervention. We construct a heuristic color similarity measure in HSV space designed to capture geometric distance from a target concept $\boldsymbol{v}$ (e.g., pure red) for an input $\boldsymbol{x}$.

First, we define the **hue similarity**. We calculate the shortest circular distance $\delta_h$ between the input hue $\boldsymbol{x}_h$ and concept hue $\boldsymbol{v}_h$. We then apply a linear decay, such that colors within $90°$ are considered similar, decreasing to zero similarity at $90°$ separation:

$$\delta_h(\boldsymbol{v}, \boldsymbol{x}) = 360° \cdot \min(|\boldsymbol{x}_h - \boldsymbol{v}_h|, 1 - |\boldsymbol{x}_h - \boldsymbol{v}_h|) \tag{27}$$

$$\text{sim}_{\text{hue}}(\boldsymbol{v}, \boldsymbol{x}) = \max\left(0°, \frac{90° - \delta_h(\boldsymbol{v}, \boldsymbol{x})}{90°}\right) \tag{28}$$

Second, we account for **vibrancy**. Hue is only meaningful when a color is vibrant (saturated and bright). For achromatic colors (low saturation or value), hue differences are irrelevant. We define "vibrancy" as the product of saturation and value, and compute the average vibrancy $r$ of the input and target:[4]

$$\text{vibrancy}(\boldsymbol{w}) = \boldsymbol{w}_s\ \boldsymbol{w}_v \tag{29}$$

$$r = \frac{\text{vibrancy}(\boldsymbol{x}) + \text{vibrancy}(\boldsymbol{v})}{2} \tag{30}$$

Third, we compute the **vibrancy-weighted hue similarity** $\widetilde{\text{sim}}_{\text{hue}}$. This term interpolates between the raw hue similarity (when colors are vibrant) and perfect similarity (when colors are achromatic/low vibrancy, making hue irrelevant):

$$\widetilde{\text{sim}}_{\text{hue}}(\boldsymbol{v}, \boldsymbol{x}) = r \cdot \text{sim}_{\text{hue}}(\boldsymbol{v}, \boldsymbol{x}) + (1 - r) \tag{31}$$

Finally, the **total similarity** combines the weighted hue similarity with proximity in saturation and value. We use linear decay terms for saturation and value differences:

$$\text{sim}(\boldsymbol{v}, \boldsymbol{x}) = \widetilde{\text{sim}}_{\text{hue}}(\boldsymbol{v}, \boldsymbol{x})(1 - |\boldsymbol{x}_s - \boldsymbol{v}_s|)(1 - |\boldsymbol{x}_v - \boldsymbol{v}_v|) \tag{32}$$

In our experiments, we instantiate the target concept $\boldsymbol{v}$ as pure red ($h = 0, s = 1, v = 1$).

**Expected Relationship Between Similarity and Reconstruction Error**    We expect reconstruction error under intervention to correlate with the square of this similarity measure. The reasoning proceeds as:

**Step 1: Latent perturbation magnitude.** The suppression intervention removes the component of latent activations aligned with the *red* anchor direction. For unit-normalized latent activations $\hat{\boldsymbol{z}} \in \mathbb{R}^E$ and concept vector $\hat{\boldsymbol{v}} \in \mathbb{R}^E$ with $\|\hat{\boldsymbol{z}}\|_2 = \|\hat{\boldsymbol{v}}\|_2 = 1$, the perturbation is:

$$\Delta\hat{\boldsymbol{z}} = -(\hat{\boldsymbol{z}} \cdot \hat{\boldsymbol{v}})\hat{\boldsymbol{v}} \tag{33}$$

with magnitude $\|\Delta\hat{\boldsymbol{z}}\| = |\hat{\boldsymbol{z}} \cdot \hat{\boldsymbol{v}}|$, which is simply the cosine similarity between the activation and the anchor direction.

**Step 2: Latent-to-output mapping.** Assuming the decoder is approximately linear locally (a reasonable assumption given the smooth, low-dimensional nature of color space), the output perturbation is:

$$\Delta\hat{\boldsymbol{y}} \approx \mathbf{J}_D \Delta\hat{\boldsymbol{z}} \tag{34}$$

where $\mathbf{J}_D$ is the decoder's Jacobian. Since the Jacobian is approximately constant in a local neighborhood, this implies $\|\Delta\hat{\boldsymbol{y}}\| \propto \|\Delta\hat{\boldsymbol{z}}\|$.

**Step 3: Quadratic error scaling.** Since reconstruction error is measured as mean squared error (MSE), we have:

$$\text{MSE} = \|\Delta\hat{\boldsymbol{y}}\|^2 \propto \|\Delta\hat{\boldsymbol{z}}\|^2 \tag{35}$$

**Step 4: Input-latent correspondence.** The key empirical assumption is that our training procedure—which anchors *red* to a predetermined direction in latent space—induces a correspondence between input-space similarity and latent-space alignment. Specifically, colors that are similar to *red* in HSV space should produce latent activations aligned with the *red* anchor direction:

$$|\hat{\boldsymbol{z}} \cdot \hat{\boldsymbol{v}}| \approx \text{sim}(\boldsymbol{v}, \boldsymbol{x}) \tag{36}$$

This is not a mathematical identity but rather an empirical relationship induced by training. The success of Sparse Concept Anchoring depends on establishing this correspondence: the structural and organizational regularizers encourage the model to align conceptually-similar inputs along consistent directions in latent space.

**Combining these steps**, we arrive at the expected relationship:

$$\text{MSE} \propto \|\Delta\hat{\boldsymbol{z}}\|^2 \propto (\hat{\boldsymbol{z}} \cdot \hat{\boldsymbol{v}})^2 \approx \text{sim}(\boldsymbol{v}, \boldsymbol{x})^2 \tag{37}$$

---

[4]We use the symbol $r$ because vibrancy is the distance from the central black-white axis in HSV, and is thus analogous to radius in a cylindrical coordinate system.

We validate this relationship empirically by computing the Pearson correlation between $\text{sim}(v, x)^2$ and reconstruction error across all test colors. Strong correlation ($R^2 \approx 0.98$) confirms that: (1) the decoder exhibits approximately local linearity, (2) the training successfully established the desired input-latent correspondence, and (3) interventions operate predictably according to the geometric structure we designed.

For permanent removal interventions (ablation), we observe a cubic relationship ($\text{MSE} \propto \text{sim}(v, x)^3$). We hypothesize this higher-order relationship arises because weight ablation affects pre-normalization embeddings $z$, altering the projection onto the hypersphere in a more complex manner than the direct subtraction of the suppression intervention (which operates on post-normalization embeddings $\hat{z}$).

## C.4    Supplementary Details for Main Experiments

Here we present regularization configurations, selection criteria, variance analysis, and scatter plots illustrating the relationship between reconstruction error and color similarity for the suppression and weight ablation experiments described in section 3. These details provide insight into model selection, robustness to initialization, and the selectivity of concept interventions.

**Anchored Architecture Regularization.**    The anchored architecture of section 3.2 uses attraction regularizers to draw labeled samples toward their target directions:

$$\mathcal{L}_{\text{concept}}(\cdot) = \lambda_{\text{anchor}} \, \Omega_{\text{anchor}}(\hat{z}, \hat{v}_{\text{red}}) + \lambda_{\text{subspace}} \, \Omega_{\text{subspace}}(\hat{z}, \mathcal{D}_{\text{vibrant}}) \tag{38}$$

The anchor term attracts *red*-labeled samples toward $\hat{v}_{\text{red}} = (1, 0, 0, 0)$, and the subspace term constrains *vibrant*-labeled samples to dimensions $\mathcal{D}_{\text{vibrant}} = \{1, 2\}$.

**Suppression.**    Figure 11 shows plots of intervention selectivity, reconstruction loss, and organization loss for the suppression experiment of section 3.2 across 60 training runs. The architecture used in these experiments was a 4-dimensional autoencoder with anchor and subspace regularization. Intervention selectivity was computed as $R^2$ between post-suppression reconstruction error $\text{MSE}(x, \hat{y})$ and the squared similarity $\text{sim}(v_{\text{red}}, x)^2$, as defined in section C.3. This quantifies how predictably the suppression intervention affects colors based on their similarity to *red*. The reconstruction loss reflects model performance on its primary objective, and organization loss reflects the overall conformance to the desired latent space structure.

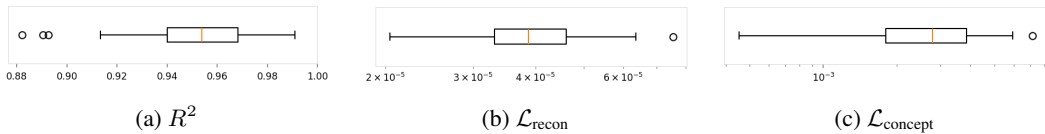

| (a) $R^2$ | (b) $\mathcal{L}_{\text{recon}}$ | (c) $\mathcal{L}_{\text{concept}}$ |

Figure 11: **Selection criteria distributions for suppression experiments.** *a*: Intervention selectivity, *b*: Reconstruction loss, and *c*: Organization loss across 60 training runs.

This architecture showed low variance across all three metrics, suggesting that the method is robust to parameter initialization. From these 60 runs, we selected the model with the highest $R^2$. Figure 12 presents scatter plots of reconstruction error versus similarity to *red* in that model, illustrating the strong quadratic relationship ($R^2 = 0.99$) achieved by suppression. This confirms that the intervention selectively increases reconstruction error for colors similar to *red*, while preserving accuracy for orthogonal colors. Weight ablation exhibits a very weak relationship in this model due to unintended selection of *anti-red* colors.

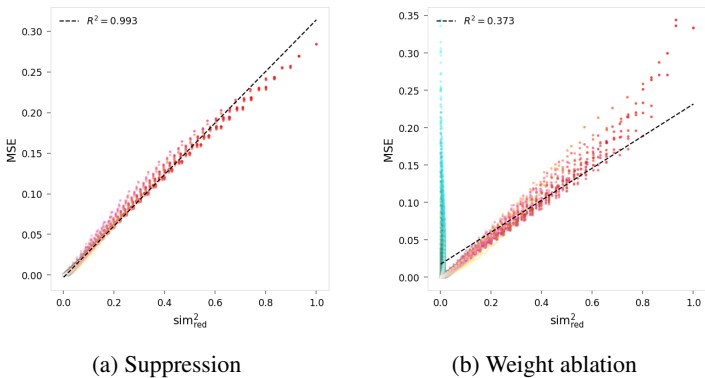

(a) Suppression          (b) Weight ablation

Figure 12: **Reconstruction error vs. similarity, anchored model. a**: Suppression shows strong quadratic relationship ($R^2 = 0.99$). **b**: Weight ablation shows poor correlation ($R^2 = 0.37$) due to the unintended selection of *anti-red* colors, visible as a vertical cluster of perturbed cyan points near $\text{sim}_{\text{red}}^2 = 0$.

**Isolated Architecture Regularization.** The isolated architecture of section 3.3 adds repulsion regularizers to reserve the anchored dimension exclusively for the target concept:

$$\mathcal{L}_{\text{concept}}(\cdot) = \lambda_{\text{anchor}}\, \Omega_{\text{anchor}}(\hat{z}, \hat{v}_{\text{red}}) + \lambda_{\overline{\text{subspace}}}\, \Omega_{\overline{\text{subspace}}}(\hat{z}, \mathcal{D}_{\text{red}}) + \lambda_{\overline{\text{anchor}}}\, \Omega_{\overline{\text{anchor}}}(\hat{z}, -\hat{v}_{\text{red}}) \quad (39)$$

The anchor term attracts *red*-labeled samples, while the two repulsion terms push all samples away from the *red* dimension ($\mathcal{D}_{\text{red}} = \{1\}$) and the *anti-red* direction ($-\hat{v}_{\text{red}}$), respectively.

**Weight Ablation.** Figure 13 presents results for the weight ablation experiment of section 3.3, across 60 training runs. The architecture was a 5-dimensional autoencoder with anchor, anti-anchor, and anti-subspace regularization as defined above. Intervention selectivity was computed as $R^2$ between post-weight-ablation reconstruction error and the cubed similarity $\text{sim}(v_{\text{red}}, x)^3$. Reconstruction and organization losses were computed as before.

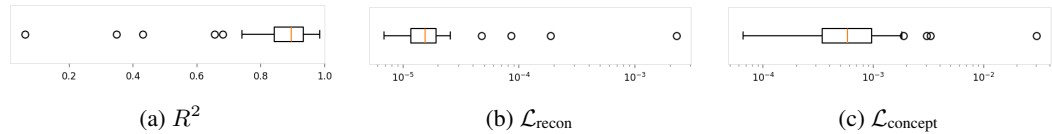

(a) $R^2$          (b) $\mathcal{L}_{\text{recon}}$          (c) $\mathcal{L}_{\text{concept}}$

Figure 13: **Selection criteria distributions for weight ablation experiments.** *a*: Intervention selectivity, *b*: Reconstruction loss, and *c*: Organization loss; across 60 training runs.

This architecture showed high variance across all three metrics, indicating sensitivity to parameter initialization. Again we selected the model with the highest $R^2$. Figure 14 presents scatter plots of reconstruction error versus similarity to *red* in that model, illustrating that weight ablation now shows a strong cubic relationship ($R^2 = 0.98$), confirming that the intervention's impact varies predictably with concept alignment.

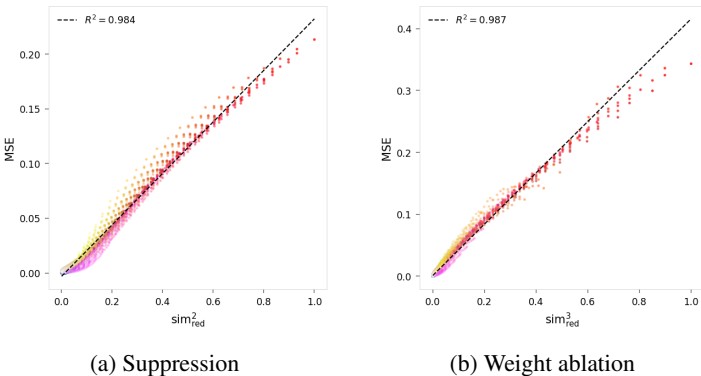

(a) Suppression        (b) Weight ablation

Figure 14: **Reconstruction error vs. similarity, with anti-subspace regularization.** *a*: Suppression retains quadratic relationship ($R^2 = 0.98$). *b*: Weight ablation shows strong cubic relationship ($R^2 = 0.98$).

## C.5 SUPPLEMENTARY EXPERIMENTS

This section presents two additional experiments that explore concept interventions under alternative organizational constraints. The first experiment demonstrates that interventions remain effective with minimal regularization (single anchor constraint). The second experiment extends our approach to multidimensional concepts, demonstrating practical control over concept subspaces.

### C.5.1 SUPPRESSION OF RED WITHOUT VIBRANT ORGANIZATION

In this experiment, we investigate the suppression of *red* in the absence of a *vibrant* organization. We aim to understand whether interventions are effective when only a single organizational regularizer is applied.

The encoder and decoder each had one hidden layer with 16 units. Latent space was four-dimensional ($E = 4$), with *red* anchored at $\hat{v}_{\text{red}} = (1, 0, 0, 0)$:

$$\mathcal{L}_{\text{concept}}(\cdot) = \lambda_{\text{anchor}} \; \Omega_{\text{anchor}}(\hat{z}, \hat{v}_{\text{red}}) \tag{40}$$

In contrast to the suppression experiment in section 3.2, no *vibrant* regularizer was used—but we expect the intervention to be similarly effective, since *vibrant* was included only for ease of interpretability.

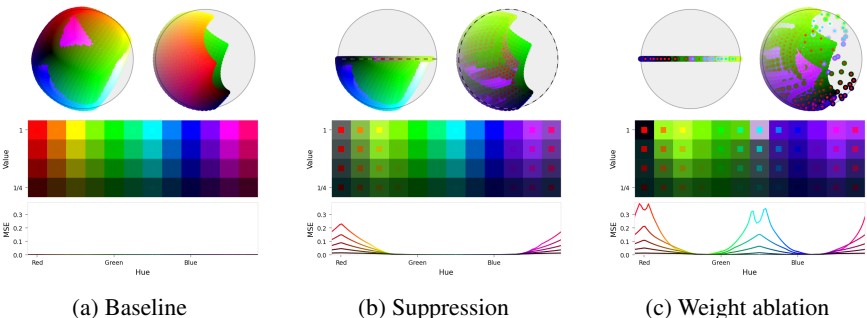

(a) Baseline        (b) Suppression        (c) Weight ablation

Figure 15: **Concept interventions with a single organizational regularizer.** A 4-dimensional autoencoder with only *red* anchored (no *vibrant* constraint). **(a)** The model structures latent space with *red* anchored as specified. **(b)** Suppression selectively increases error for *red* while preserving other colors. **(c)** Weight ablation increases error for both *red* and *cyan*.

The results are consistent with previous findings: suppression of *red* increased reconstruction error specifically for *red* colors while preserving reconstruction quality for other colors (see figure 15 and table 2). Weight ablation again increased error for both *red* and its opposing color, *cyan*. These

results indicate that concept interventions can be effective even when only a single organizational regularizer is applied.

Table 2: **Suppression selectively targets *red* in a single-constraint model.** Reconstruction error (MSE) for baseline and suppression across representative hues, values, and achromatic colors. Suppression increases error for *red* while preserving reconstruction quality for orthogonal colors. Weight ablation also affects opposing colors.

| Color | | Baseline | Suppression | Weight Ablation |
|---|---|---|---|---|
| Red | ■ | 0.00 | +0.23 | +0.33 |
| Lime | ■ | 0.00 | +0.00 | +0.00 |
| Cyan | ■ | 0.00 | +0.00 | +0.25 |
| Purple | ■ | 0.00 | +0.00 | +0.00 |
| Black | ■ | 0.00 | +0.00 | +0.00 |
| Gray | ■ | 0.00 | +0.00 | +0.00 |
| White | □ | 0.00 | +0.00 | +0.00 |

### C.5.2   WEIGHT ABLATION OF HUE SUBSPACE

In this experiment, we perform weight ablation of the entire *hue* subspace to test our method on multidimensional concepts.

The encoder and decoder each had one hidden layer with 16 units. Latent space was four-dimensional ($E = 4$), with *vibrant* confined to dimensions $\mathcal{D}_{\text{vibrant}} = \{1, 2\}$:

$$\mathcal{L}_{\text{concept}}(\cdot) = \lambda_{\text{subspace}} \, \Omega_{\text{subspace}}(\hat{z}, \mathcal{D}_{\text{vibrant}}) \tag{41}$$

After training, we ablate all weights connected to the *hue* subspace (dimensions 1 and 2) according to equation (15). To allow comparison with earlier experiments, we also implement a suppression intervention targeting the *vibrant* subspace:

$$\hat{z}' = \hat{z} - P_{\mathcal{D}_{\text{vibrant}}}(\hat{z}) \tag{42}$$

where $P_{\mathcal{D}_{\text{vibrant}}}(\hat{z})$ is the projection of $\hat{z}$ onto the subspace spanned by dimensions in $\mathcal{D}_{\text{vibrant}}$, effectively zeroing out those dimensions. Like directional suppression—but unlike weight ablation—this is applied post-normalization, resulting in activation vectors of length $|\hat{z}'| \leq 1$.

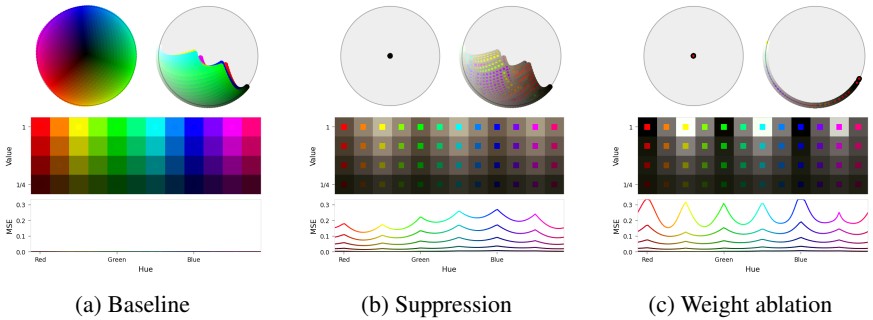

|   (a) Baseline   |   (b) Suppression   |   (c) Weight ablation   |

Figure 16: **Deletion of a multidimensional subspace.** A 4-dimensional autoencoder with *vibrant* colors confined to a 2D subspace (no *red* constraint). **(a)** The model organizes *vibrant* colors in dimensions 1–2, with achromatic colors occupying orthogonal dimensions. **(b)** Suppression of the *vibrant* subspace removes hue information, mapping all colors toward achromatic values while preserving brightness. **(c)** Weight ablation produces similar results, degrading reconstruction of all chromatic colors.

The results demonstrate that Sparse Concept Anchoring extends naturally to multidimensional concepts. As shown in figure 16 and table 3, both suppression and weight ablation of the *vibrant* subspace substantially increase reconstruction error for all chromatic colors (red through magenta), while reconstruction of achromatic colors (black through white) remains largely unaffected. This indicates that the model successfully organized *vibrant* information within the specified subspace, with achromatic information occupying orthogonal dimensions.

Table 3: **Targeted degradation of chromatic colors.** Reconstruction error (MSE) for baseline, suppression, and weight ablation across representative hues and achromatic colors. Both suppression and weight ablation of the *vibrant* subspace increase error for all chromatic colors while preserving achromatic reconstruction.

| Color | | Baseline | Suppression | Weight Ablation |
|---|---|---|---|---|
| Red | | 0.00 | +0.18 | +0.33 |
| Lime | | 0.00 | +0.14 | +0.16 |
| Cyan | | 0.00 | +0.26 | +0.31 |
| Purple | | 0.00 | +0.20 | +0.19 |
| Black | | 0.00 | +0.00 | +0.00 |
| Gray | | 0.00 | +0.00 | +0.00 |
| White | | 0.00 | +0.00 | +0.00 |

Unlike single-direction interventions (e.g., sections 3.2 and 3.3), subspace interventions target manifold-structured concepts that cannot be captured by a single direction. The circular organization of hues in the *vibrant* subspace exemplifies such structure—analogous to cyclical concepts like days of the week or temporal patterns in language models. Both suppression and weight ablation successfully eliminate this circular structure, demonstrating that our approach applies to concepts with non-trivial geometric organization.

An interesting pattern emerges in the reconstructions: although MSE is similar across chromatic colors, the resulting brightness varies systematically by hue. Red, green, and blue become darker, while yellow, cyan, and magenta become brighter (figures 16b and 16c). This reflects the geometry of the RGB cube (section B.5): when hue information is removed, colors collapse toward the achromatic axis (the black-white diagonal). Primary hues (red, green, blue) lie closer to black, while secondary hues (yellow, cyan, magenta) lie closer to white along this axis. The systematic brightness variation thus arises from the geometric structure of RGB space rather than any bias in the intervention method.

## D    EXPERIMENTAL DETAILS

All experimental details can be found in the code at https://github.com/z0u/ex-preppy/tree/a91164f. Results are best viewed at https://z0u.github.io/ex-preppy.

The primary experiments correspond to the following sections:

- **Experiment 2.4.1**: Anchored architecture for suppression (section 3.2)
- **Experiment 2.9.1**: Isolated architecture for weight ablation (section 3.3)
- **Experiment 2.5.1**: Suppression of red without vibrant organization (section C.5.1)
- **Experiment 2.7.1**: Weight ablation of hue subspace (section C.5.2)

## E    THE USE OF LARGE LANGUAGE MODELS (LLMs)

We acknowledge the use of Large Language Models (LLMs) in the creation of this work. LLMs were employed to refine our initial research idea through discussion of conceptual gaps and potential solutions, accelerate the development of methods and experiments using coding assistance, and support literature review by identifying relevant prior work. Additionally, LLMs contributed to improving the clarity, grammar, and overall flow of the text. For these purposes, we used OpenAI's ChatGPT and Anthropic's Claude family for ideation, literature discovery, and writing, and GitHub's Copilot for coding assistance (various models).

