# OpenReview forum: "Sparse Concept Anchoring for Interpretable and Controllable Neural Representations"
_ICLR.cc/2026/Workshop/GRaM — ICLR 2026 Workshop GRaM Poster_

### Official Review · Reviewer_MPaL · 2026-02-15
**Sparse Anchoring for Interpretability through Minimal Concept Supervision**

**Rating:** 7
**Confidence:** 3

**Review:**

## Strengths

- **Relevant idea, well contextualized and introduced**: The paper addresses a relevant concept in AI safety: the trade-off between post-hoc discovery and full concept supervision. The idea seems original, shifting the focus to an "interpretability-by-design" framework. By anchoring concepts to known geometric coordinates during training, the authors provide a way of preserving model flexibility while ensuring that safety-critical features remain accessible for intervention. This approach is well-situated within the broader context of representation learning, effectively bridging the gap between unsupervised feature emergence and the rigid constraints of Concept Bottleneck Models.

- **Method is clearly described**: The method is presented with clarity, with the distinction between Suppression (activation-level) and Ablation (weight-level) being well established. Authors also detail how to perform interventions, and the use of a hypersphere ($\mathbb{S}^{E-1}$) for geometric consistency is well-justified.

- **Thorough experiment on color autoencoders**: Using the RGB cube provides a "ground truth" geometry that makes the the anchoring intuitive and visually verifiable. Achieving selective removal with labels for only < 0.1% of data is a significant empirical result.

- **Extensive discussion of limitations and future work**: The authors address limitations of the current model, including the "adversarial recovery" risk and the challenges of scaling to linguistic manifolds, which is in line with some of the weaknesses below)

## Weaknesses

- **Limited generalization**: While RGB is a good example to test the method, it is a low-dimensional, continuous, and highly structured space. It remains unclear if SCA can handle the polysemanticity found in high-dimensional LLM latent spaces, where the concepts are rarely perfectly orthogonal. Further applications on this and other real-world scenarios could strengthen the contributions.

- **Scalability of constraints**: As the number of anchored concepts ($K$) increases, the cumulative effect of multiple repulsion and attraction terms ($\lambda_{sep}, \lambda_{ck}$) may create a difficult optimization landscape. It would be interesting to explore the point where anchoring might start to degrade the primary task performance ($L_{task}$).

- **Model complexity**: The experiments rely on shallow autoencoders (1–2 hidden layers) while modern architectures (Transformers) rely on residual streams where information is added iteratively. It would be interesting to explore if a single anchor at one layer would be bypassed by the model in deeper layers.

- **Presentation of experimental setup**:  Architecture details and results seem a bit mixed in Sections 3.2/3.3, which makes it difficult for the reader to follow the experimentation. A better separation would help improve clarity.

- **Selection bias in "best" model**: The authors select the "most selective" model from 60 runs using a Pareto frontier. While transparently reported, this suggests the method may be sensitive to seed initialization, potentially requiring many restarts to achieve the desired geometric isolation.

**Pmlr Suitability:**

Yes

---

### Official Review · Reviewer_Egtk · 2026-02-18
**Offical GRaM Submission19 Review**

**Rating:** 7
**Confidence:** 3

**Review:**

Overview: This long paper proposes a geometry-first approach to interpretability by anchoring selected concepts at predetermined locations in latent space using minimal supervision, enabling reliable behavioral steering and targeted concept removal. Through controlled experiments on color autoencoders, the research demonstrates that explicit geometric constraints can yield highly selective and predictable interventions.

Strengths:

-The paper presents a geometry-first framework that anchors concepts at predetermined locations in latent space, enabling predictable and reliable interventions.

-Explicit geometric constraints on the hypersphere provide a principled foundation for both reversible and behavioral steering and permanent concept removal.

-Experimental results show strong selectivity, with targeted concepts modified while orthogonal features remain largely unaffected.

Weaknesses:

-Experimental validation is limited to low-dimensional color autoencoders, which can be improved by addressing scalability to high-dimensional and real-world domains.

-While the geometric interventions are effective, the security and robustness implications of cleanly isolated concept dimensions are not fully explored.

**Pmlr Suitability:**

Yes

---

### Official Review · Reviewer_yHhV · 2026-02-20
**An Elegant Geometric Approach, but Requires Better Contextualization of Results and Scalability**

**Rating:** 6
**Confidence:** 3

**Review:**

**Summary*
This paper proposes "Sparse Concept Anchoring," a method to establish predictable, interpretable latent spaces by combining global structural constraints (hypersphere normalization and separation regularizers) with sparse concept-organizational regularizers. By anchoring a small fraction of labeled data (<0.1%) to specific dimensions, the authors demonstrate that they can control the latent geometry well enough to perform targeted interventions: either dynamic behavioral steering (suppression) or permanent concept removal (weight ablation). The method is evaluated on a controlled RGB color autoencoder dataset.

**Strengths**

*1. Important Problem & Novel Approach:* The paper addresses a critical challenge in AI safety and interpretability—intervening in latent spaces reliably. The shift from post-hoc discovery of steering vectors to a priori geometric anchoring during training is a highly promising direction.

*2. Sound Mathematical Foundation:* The blend of structural constraints to enforce global geometry and targeted regularizers makes intuitive and mathematical practical sense.

*3. Clarity of Background:* The method section is generally well-written, and the inclusion of the conceptual schematics in Appendix B.2 provides excellent intuition for the geometric forces at play.

**Weaknesses & Questions**
While I lean favorably toward the core idea, the manuscript requires streamlining in its presentation of results, and the claims regarding supervision sparsity need to be appropriately scoped.

*1. Overclaiming the "<0.1% Data" Metric:* The abstract and introduction heavily emphasize the ability to anchor concepts using labels for "<0.1% of examples." While impressive in this specific context, this metric is heavily dependent on the extreme simplicity and low dimensionality of the toy RGB dataset (where concepts like "red" form a clean, highly structured basis).

* Feedback: There is currently no heuristic or discussion indicating how this minimal supervision requirement scales to higher-dimensional, complex manifolds (e.g., semantic concepts in LLMs or complex vision tasks). While state-of-the-art results are not required for a workshop paper, the authors should temper this claim in the abstract and add a discussion (perhaps in the limitations or appendix given size constraints) addressing how the required fraction of labeled data might scale with dataset complexity or manifold curvature.

*2. Interpretation of Figures and Results:* The presentation of the results in Section 3 assumes a high degree of familiarity with this specific type of color-space evaluation. Figures 2 and 3 are dense and lack sufficient textual hand-holding. While it is visually apparent that the "red" concept is removed in Figure 1 for example, the mechanics of how to read the visual grids and expected outcomes are largely left to the reader.

* Suggestion: Add a dedicated paragraph explicitly walking the reader through how to interpret the grids and curves in the figures (e.g., contrasting the target dot with the generated square, and explaining what the specific spikes in the error curves mean in relation to the geometric intervention).

*3. Streamlining the Flow of Section 3:*
The transition between Section 3.2 (Anchored Architecture) and Section 3.3 (Isolated Architecture) is somewhat jarring. It is initially difficult to understand why both architectures are presented and how their results connect.

* Suggestion: The narrative would benefit greatly from a brief synthesizing subsection (e.g., "3.4 Intervention Trade-offs") that explicitly compares the two. It should clearly articulate that while 3.2 is sufficient for inference-time suppression, 3.3's repulsive regularizers are a necessary architectural addition to prevent collateral damage during permanent weight ablation.

**Pmlr Suitability:**

Yes

---

### Meta-Review · Area_Chair_t1bn · 2026-02-23

**Decision:**

Accept

**Metareview:**

The authors introduce Sparse Concept Anchoring to achieve interpretable latent representations. All the reviewers agree that the paper is novel, with a mix of theoretical and empirical contributions.

**Relevance To Proceedings:**

Yes — suitable for PMLR (long paper)

**Relevance To Workshop:**

Yes — suitable for GRaM

---

### Decision · Program_Chairs · 2026-03-02

Accept (Poster)